# Copolymers of 4-Trimethylsilyl Diphenyl Acetylene and 1-Trimethylsilyl-1-Propyne: Polymer Synthesis and Luminescent Property Adjustment

**DOI:** 10.3390/molecules28010027

**Published:** 2022-12-21

**Authors:** Tanxiao Shen, Manyu Chen, Haoke Zhang, Jing Zhi Sun, Ben Zhong Tang

**Affiliations:** 1MOE Key Laboratory of Macromolecular Synthesis and Functionalization, Department of Polymer Science and Engineering, Zhejiang University, Hangzhou 310027, China; 2Centre of Healthcare Materials, Shaoxing Institute, Zhejiang University, Shaoxing 312000, China; 3Hangzhou Global Scientific and Technological Innovation Center, Zhejiang University, Hangzhou 311215, China; 4Shenzhen Institute of Aggregate Science and Technology, School of Science and Engineering, The Chinese University of Hong Kong, Shenzhen 518172, China

**Keywords:** copolymerization, disubstituted-acetylene, fluorescence, AIE, explosive detection

## Abstract

Poly(4-trimethylsilyl diphenyl acetylene) (PTMSDPA) has strong fluorescence emission, but its application is limited by the effect of aggregation-caused quenching (ACQ). Copolymerization is a commonly used method to adjust the properties of polymers. Through the copolymerization of 4-trimethylsilyl diphenyl acetylene and 1-trimethylsilyl-1-propyne (TMSP), we successfully realized the conversion of PTMSDPA from ACQ to aggregation-induced emission (AIE) and aggregation-induced emission enhancement (AEE). By controlling the monomer feeding ratio and with the increase of the content of TMSDPA inserted into the copolymer, the emission peak was red-shifted, and a series of copolymers of poly(TMSDPA-*co*-TMSP) that emit blue–purple to orange–red light was obtained, and the feasibility of the application in explosive detection was verified. With picric acid (PA) as a model explosive, a super-quenching process has been observed, and the quenching constant (*K*_SV_) calculated from the Stern–Volmer equation is 24,000 M^−1^, which means that the polymer is potentially used for explosive detection.

## 1. Introduction

Polyacetylene is a well-known conjugated polymer because of the metallic conductivity found in its highly doped films, which opened a new area of research on “synthetic metals” [1,2,3,4]. However, pristine polyacetylene is very unstable and intractable, and thereby the scope of its practical applications has been heavily limited. Replacement of the two hydrogen atoms on the acetylene monomer with appropriate substituents endows disubstituted acetylenes, and the corresponding polymers or poly(disubstituted acetylene)s (PDSAs) exhibited good stability and processability [5]. In recent years, PDSAs have shown advanced functions and promising applications in some high-tech areas, such as ultra-high gas permeability [6,7,8] for the transportation of small-molecule gases and solvents and luminescent elastomers for stimuli-responsive materials [9,10,11,12,13,14]. Due to the poor tolerance of polymerization catalysts to polar functional groups, however, the disubstituted acetylene monomers available for polymerization are very limited [15], which seriously restricts the related basic research and the development of new products. From the perspective of synthetic chemistry, there are mainly three ways to change the functions of polymeric materials. One is to change the polymerization reaction route so that the functional monomer can be polymerized to obtain the desired performance. However, some polymerization routes have problems to be solved, such as a long reaction time, low conversion rate and yield, etc. The second is to change the polymerization catalyst and adjust the polymer chain structure, to obtain an improved performance. It is very difficult to develop new catalysts, and only palladium (Pd)-based catalysts [16] have been developed so far. Most PDSAs are polymerized using conventional early transition metal compounds such as WCl_6_, MoCl_5_, NbCl_5_, and TaCl_5_ as catalysts. The third is to adjust the chemical composition and chain structure of the polymers by copolymerization to achieve the purpose of property modification.

Copolymerization, as one of the most commonly used modification methods, has also been reported for the preparation of PDSAs, but it is far less than that of olefin-based copolymers. The documented reports mainly focus on the two PDSAs, poly(1-trimethylsillyl-1-propyne) (PTMSP) and poly(4-trimethylsilyl diphenyl acetylene) (PTMSDPA). Since the first report of PTMSP in 1983 [17], a series of publications has appeared due to its unique structure and properties and, in particular, the extremely high gas permeability. Hamano [18] noticed that TMSP could copolymerize with other double-substituted acetylene monomers under the catalysis of TaCl_5_/Ph_3_Bi or NbCl_5_/Ph_3_Bi, and the reaction activity sequence of monomers was obtained through the copolymerization curve: 2-octyne > TMSP > 4-octyne > 1-phenyl-1-propyne > 1-phenyl-1-hexyne. Ghisellini [19] realized the copolymerization of TMSP with 1-trimethyl-1-hexene, and the catalyst used was TaCl_5_/Ph_3_Bi. Khotimsky and colleagues [20] investigated the copolymerization of 4-methyl-2-pentyne and TMSP initiated by catalytic systems based on NbCl_5_. PTMSDPA as a very unusual π-conjugated polymer has a polyene backbone and two side phenyl rings and is quite emissive [21] and responsive to external stimuli such as liquid solvents [22]. Kwak [23] explored PTMSDPA copolymers containing both longer and shorter alkyl side chains in the same backbone.

All the above copolymerized monomers have very similar structures, so there is great potential to regulate the property of PDSAs through copolymerization. We prepared a series of copolymers of TMSP and TMSDPA, as shown in Figure 1. The characterization data confirmed that the resultant polymers possess the expected chemical structure, and the properties of these copolymers were investigated. We took advantage of the AIE properties of P**7** to detect picric acid (PA), a model compound of nitro-aromatic explosives in aqueous media, and a high sensitivity to PA has been achieved.

## 2. Results and Discussion

### 2.1. Copolymerization

Copolymerization of the two monomers was conducted by using TaCl_5_–Bu_4_Sn, which was selected from seven catalysts (TaCl_5_–Bu_4_Sn, TaCl_5_–Ph_3_SiH, NbCl_5_–Ph_4_Sn, NbCl_5_–Ph_3_SiH, WCl_6_–Ph_4_Sn, WCl_6_–Ph_3_SiH, and MoCl_5_–Bu_4_Sn), because the copolymerization reaction could only proceed under this catalyst. Table 1 presents the results of the copolymerization of the monomers. The polymer yield of P**1**, regarded as adding a small amount of TMSDPA to the TMSP system, was reduced to 50% of PTMSP, although the molecular weight was not significantly changed. Accordingly, it could be suggested from P**4** that the reaction could not succeed once TMSP was added to TMSDPA. Further research showed that when the contents of TMSDPA reached 60% and 80%, there was no product. When TMSP reached 60% and 80%, small amounts of products could be obtained. Then, we speculated that the complex catalyst of TaCl_5_–Bu_4_Sn tends to polymerize TMSP.

We also examined the time effect of the polymerization reaction. The molecular weight reached 10^6^ even though the reaction time was only 2 min for the homo-polymerization of TMSP and TMSDPA. Gel was produced when the reaction time was extended to 10 min. The results indicate that the catalyst has very high activity in the homo-polymerization of TMSP and TMSDPA. The product yield improved with the increase of the reaction time, but the molecular weight was almost unchanged, as suggested by the comparison between P**5** and P**9**. These results indicate that the initiation of the active center is the rate-determining step in polymerization, and the chain growth rate is extremely high. The results of P**10** and P**11** show that the yield and molecular weight of the polymer were not significantly affected by changing monomer sequences.

### 2.2. Structural Characterization

FTIR and ^1^H NMR spectroscopic techniques were employed to confirm the chemical structure of the polymers. We used the characterization data of P**7** as representative of the copolymer to compare with the homopolymer. Other copolymers have similar data (Appendix A). As clearly shown in the FTIR spectra (Figure 1), the absorption band around 2200 cm^−1^ that could be attributed to the tensile vibration of the C≡C bond was absent. At the same time, a new asymmetric absorption band near 1600 cm^−1^ appeared, which can be assigned to the asymmetric stretching vibration peak of the C=C bond. The changes of these two characteristic bands indicate the complete consumption of the triple bond and transformation into the double bond after the polymerization reaction. Infrared absorption spectra were used to further characterize the influence of the reaction time on the polymer composition (Appendix A). The overall infrared spectrum changes showed a trend of superposition of PTMSP and PTMSDPA spectra with the reaction time. The absorption band of 1120 cm^−1^ assigned as the respiratory vibration of the benzene ring appeared in the infrared spectrum since P**8**, indicating that the TMSDPA component in the copolymer increased with the increase of time. 

Further evidence comes from the ^1^H NMR spectra (Figure 2) of PTMSP, P**7**, and PTMSDPA. The signal in the low-field part of the spectrum was amplified for analysis. The resonant peak around 7.5 ppm, which was assigned to the protons of the benzene ring, existed in P**7**. The monomer showed a sharp resonance signal around 0.1 ppm, which was assigned to the protons of methyl and transformed into a dull band. The above results suggest that copolymerization of phenyl disubstituted acetylene and alkyl-silyl disubstituted acetylene has been successfully achieved. 

### 2.3. Photophysical Property

Poly-diphenyl acetylenes are highly emissive even in a solid state since the introduction of the benzene ring changes the electronic structure of the backbone, and the fluorescence property of PTMSDPA has been widely reported [24,25,26,27,28]. Since no one has tried to synthesize copolymers containing PTMSDPA chain segments, the absorption and emission properties of these copolymers have never been reported anywhere. Figure 3a shows the UV-visible absorption spectra of copolymers. 

A straight line in the range from 370 to 620 nm of PTMSP was recorded, and this observation is consistent with the appearance of the white powder. This means that the polymer does not have the extended electronic structure of general polyacetylenes, because the large substituents distort the conjugation of the polyene backbone. PTMSDPA has two obvious absorption peaks at 430 and 375 nm, respectively. The absorption band of 430 nm can be assigned to the π-π* transition of the conjugated backbone of the polymer, and the absorption band of 375 nm can be assigned to the π-π* transition of the diphenylethylene conjugated unit. There are no characteristic absorption peaks of 375 nm and 430 nm assigned to PTMSDPA in P**1** and P**2**, indicating that P**1** and P**2** contain few or no diphenylethylene conjugate units, and this is consistent with the results of structural characterization. P**5** and P**6** still do not show the characteristic absorption band at 430 nm, suggesting that it is difficult to link the TMSDPA monomer to the PTMSP chain end when the reaction time is short. P**7** shows the character of PTMSDPA’s UV absorption peak. This indicates that when the reaction time is prolonged to a sufficient degree, the TMSP monomer is consumed and the active center begins to accept the introduction of TMSDPA, resulting in the occurrence of PTMSDPA fragments in the polymer chain. As a result, the UV-visible absorption spectrum of the polymer shows the characteristic absorption band of its homopolymer. The UV-visible absorption spectra of the polymer P**8** and P**9** were more obvious by prolonging the polymerization time to 30 and 50 min, which indicates that there are already enough PTMSDPA segments in the polymers. Compared with P**10**, the stronger UV absorption peak of P**11** (Appendix A) at 415 nm indicates that there are more TMSDPA fragments in the main chain, which further indicates that it is difficult for TMSDPA to insert into the PTMSP chain.

The above deduction was supported by the experimental data shown in Figure 4. The PTMSP was non-emissive in THF solution, and a strong luminescence peak of 498 nm appeared in the photoluminescence (PL) spectrum of PTMSDPA. The PL spectrum of P**1** almost overlaps with PTMSP. However, P**2** shows a weak luminescence peak at around 490 nm, indicating that the copolymerization was successful. This is inconsistent with the previous results because fluorescence spectra are more sensitive than FTIR, ^1^H NMR, and UV-visible absorption spectra. P**7**–P**9** solutions showed gradually enhanced fluorescence emission. The red-shift fluorescence emission peak of P**11** (Appendix A) compared with P**10** confirmed our previous results. The luminescent performance of P**5** and P**6** looks abnormal: their dilute solutions have not shown absorption and emission bands, which implies that there are no TMSDPA segments in polymers. However, their powders emit fluorescence under UV light (Appendix A). This phenomenon can be associated with the aggregation-induced emission (AIE) effect. For a typical AIE molecule or macromolecule, there is no or very weak luminescent emission observed for the dilute solution of a good solvent, however significantly enhanced luminescent emission can be recorded when the molecules form aggregates or are trapped in a certain cramped environment [29,30,31,32]. Tetraphenylethylene (TPE) is a representative AIE-active luminogen (AIEgen), and the structure for the main chain of poly(TMSP-co-TMSDPA) containing two adjacently inserted TMSDPA units is similar to that of TPE. The absence of fluorescence emission of P**6** in the solution state does not mean the absence of TMSDPA units in the polymer chain but it indicates that there are very short segments of TMSDPA in the polymer chain, serving as the source of the AIE property.

The fluorescence spectra (Figure 5) of solid powders of polymers P**5**–P**9** were measured to verify the above deduction. The solid powders of P**5** and P**6** both emitted blue fluorescence with a peak at around 470 nm when a larger ratio of TMSDPA monomer was introduced into the polymer chain with the extension of the polymerization time, and the PTMSDPA segments were lengthened. Consequently, the emission spectrum was red-shifted. The emission peaks of P**7**–P**9** solids appeared at 545, 555, and 558 nm, respectively. According to the classical method to verify the AIE properties of molecules, the PL behavior of the polymers in the mixed solvent system of THF/H_2_O with different water contents was analyzed by using THF as a good solvent and water as a bad solvent (Figure 6, Appendix A). The excitation wavelength of 375 nm was chosen to avoid the effect of Raman scattering by water. Figure 6 shows that with the increase of the water fraction in the mixed solvent, the system changed from non-luminescent to luminous, and the fluorescence intensity increased with the increase of the water content, which fully confirms that P**6** is an AIE-type polymer. There was weak fluorescence in the P**7** solution. The fluorescence was gradually enhanced when water, the poor solvent of P**7**, was added to the solution, and the solution reached the highest fluorescence intensity when the water fraction reached 90%. The polymer exhibits typical AEE behavior.

On the other hand, PTMSDPA shows a typical ACQ behavior (Appendix A). The fluorescence intensity decreased with the increase of water fraction, which is similar to the luminescence behavior of most diphenyl acetylene derivatives [21,22,23,24,25]. According to the above results, a series of copolymers containing different lengths of PTMSDPA can be obtained by controlling the proportion of the two monomers and the reaction time. We can not only regulate the luminescence color from blue–purple to orange–red, but we can also adjust the luminescent behavior from ACQ of the homopolymer PTMSDPA to AIE of the copolymer.

### 2.4. Explosive Detection

Detection of explosives has drawn much attention because of its application in global safety and anti-terrorism activities. In this area, fluorescence spectroscopy has been widely used due to its high sensitivity, good selectivity, low cost, and easy operation. The suspension formed by P**7** in the mixed solvent shows high stability compared with the suspension formed by small molecules and can be placed for several months without precipitation, which makes the system more suitable for working as a detection medium. Then, P**7** in the THF/water mixture with a 90% water fraction was used as the fluorescent detection medium, and picric acid (PA, 2,4,6-trinitrophenol) was chosen as a model of explosive to simulate practical detection. As shown in Figure 7a, the intensity of the fluorescence began to decrease with the addition of more and more PA into the suspension, but the maximum emission wavelength did not change. In the PA concentration range of 0~100 μM, an exponential growth (*I_0_*/*I* vs. PA concentration) can be derived (Figure 7b), which implies that the detection system has a super-amplification effect, as observed in other AIE-active polymer systems [33,34,35,36,37,38,39,40,41]. Based on the experimental data, the quenching constant (*K*_SV_) of PA to P**7** was calculated from the Stern–Volmer plot, which is as high as 24,000 M^−1^. This attempt suggests that P**7** is potentially used as an effective fluorescent probe in the fabrication of test plates and/or portable sensors for explosive detection.

## 3. Materials and Methods

### 3.1. Materials 

The 1-Trimethylsillyl-1-propyne (TMSP) and TaCl_5_ were purchased from J&K Scientific (Beijing, China). Bu_4_Sn, Ph_4_Sn, Ph_3_SiH, and Et_3_SiH were purchased from Energy Chemical (Shanghai, China). WCl_6_, MoCl_5_, and NbCl_5_ were purchased from Sigma-Aldrich Corporation (Shanghai, China). The 4-Trimethylsilyl diphenyl acetylene (TMSDPA) was purchased from Tokyo Chemical Industry (Shanghai, China). All the chemicals were used directly without further purification. Tetrahydrofuran (THF) and toluene were freshly distilled from sodium and benzophenone ketyl under nitrogen and normal pressure. 

### 3.2. Instruments 

The molecular weight (*M*_w_ and *M*_n_) and polydispersity (*M*_w_/*M*_n_) of the polymers were estimated in THF using a gel permeation chromatography (GPC, PL-GPC-50, Waters Corporation, Milford, CT, USA) system with a set of monodisperse polystyrene standards covering the molecular weight varying from 10^3^ to 10^7^ as calibration. FTIR spectra were recorded on a VECTOR 22 (Bruker Corporation, Billerica, MA, USA) spectrometer. ^1^H NMR spectra were recorded on AVANCE III 400 (Bruker Corporation, Billerica, MA, USA) spectrometers, and tetramethylsilane (TMS) was used as an internal standard. UV-vis absorption spectra were recorded on a Varian CARY 100 Bio UV-vis (Agilent Technologies Inc, Santa Clara, CA, USA)spectrophotometer. Fluorescence spectra were recorded on a RF-5301PC (Shimadzu, Kyoto, Japan) spectrofluorophotometer. Quantum yield was recorded on the Spectrofluorometer FS5 (Edinburgh Instruments, Livingston, Scotland, UK) spectrofluorophotometer.

### 3.3. Polymer Synthesis 

The polymerizations were carried out under the protection of dry nitrogen using the standard glovebox or Schlenk technique, except for polymer purification. The catalyst (1 eq., 1.0 mmol) and co-catalysts (2 eq., 2.0 mmol) were stirred with 400 rpm in a desired solution (3 mL) at 80 °C for 20 min, and then 3 mL of toluene solution of the monomers (10 eq., 10 mmol) was added. The mixture was stirred at 80 °C for the desired period of time and the reaction was quenched with nearly 8–10 mL of methanol and washed with 120 mL of methanol and acetone three times to precipitate a solid polymer. Then, the solid sample was filtered by a filter funnel with a fritted disc and dried under a vacuum for 24 h. 

## 4. Conclusions

In this article, copolymers of disubstituted acetylenes containing both phenyl and alkyl-silyl in the same backbone were synthesized and the chemical structures of the resultant polymers were characterized with FTIR and NMR spectroscopy techniques. Their emission properties varied according to the proportion of the two monomers. A higher content of TMSDPA units in the copolymers led to significantly enhanced, red-shifted emission in the solid and solution, and polymers were adjusted from ACQ to AIE copolymers. Consequently, it was possible to finely tune the emission properties of the polymers simply via the copolymerization of monomers. Moreover, using the P**7** suspension and PA as the detection medium and model explosive, a super-quenching process has been observed and a good linear relationship between ln(*I*_0_/*I*) vs. PA has been established. The quenching constant (*K*_SV_) calculated using the Stern–Volmer equation was 24,000 M^−1^. This result is expected to be quite helpful in the molecular design of light-emitting materials with finely tuned fluorescence emission and in the further research of poly(disubstituted acetylene) derivatives.

## Data Availability

Not applicable.

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
