# Peer review of "Copolymers of 4-Trimethylsilyl Diphenyl Acetylene and 1-Trimethylsilyl-1-Propyne: Polymer Synthesis and Luminescent Property Adjustment"

_molecules, 2022, doi:10.3390/molecules28010027_

Round 1
Reviewer 1 Report
My comments are:
In the abstract part, the authors should explain the meaning of the result that the quenching constant is 24000 M-1.
You should add space before adding references.
Introuduction. Line 71. You have to add the complete name of AIE. After that, only write the initials.
Is only one arrow in scheme 1? this means that the reaction is not reversible. In the text, the authors said that the yield is lower than 100 %. So. maybe the reaction is reversible and, in that case, you have to put doble arrow. Or maybe is not reversible and you have other products. Justify it.
Which solvent do you use in the reaction? why that solvent?
The authors do not explain what is the meaning of PDI parameter and which is the effect caused in the obtained product.
Table 1 shows that the longer the reaction time, the higher the yield is obtained. Why then do they not use times greater than 50 minutes?
In figure 1, on the x-axis you must put a space between the wave number and the units.
In section 2.3 (photophysical properties), line 140. Sample 4 does not appear in figure 3. It must be an error.
The scales in Figures 3a and 3b are different. Please adjust them. In these figures, on the x-axis you must put a space between the wave number and the units again.
In section 2.3 (photophysical properties), line 172. There is a mistake in the word "fluoresce". Correc it, please.
Does the pH of the reaction solution influence? if so, what would happen at different pH's should be studied.
Why do you study the concentration range from 0 to 100 micrometers? What happens at lower o higher concentrations?
In section 2.4 (explosive detection), line 235. Remove the space in 24000 M-1
There is a mistake in figure 7b. You should add a space between [TNP] and the units.
In section 3.3 (polymer synthesis). Which filter do you use to filtrate the products? Which are the rpm when you stirring? and how many time do you dried the final product under vacuum?
Author Response
please find our responses in the attached file. Thanks!

Reviewer 2 Report
This is a well-written paper that reports a series of polymerized materials that have a unique AIE phenomenon. By controlling the copolymerization of monomers, the resulting copolymer displays tunable emission properties. The finding is interesting, scientifically sound, and well performed. The manuscript is publishable after extensive proofreading for grammar and style.
1. In the abstract, the author should define the’ACQ, AIE and AEE’ since these short names showed up for the first time.
2. Compound numbers should be systematically in bold within the manuscript, including in tables, figures, and their caption.
3. For the Stern–Volmer plot, the I0/I line is curved, which may indicate there has some dynamic quenching or even more complicated mechanism happening, thus the so-called ‘super-amplification effect’ should be more specifically explained.
4. The quantum yield for the AIE suspension and other samples should be given.
The authors should carefully read the grammatical errors and inappropriate scientific language throughout the text:
For instance, Line 70: ‘… We took into account the advantages of…’ should be ‘…we took advantage of …’ if I was right.
Line 229, ’more and more’ is too casual
The title needed to reconsider.
Author Response
Please find our responses in the attached file. Thanks!
